# Comparison of Biocompatibility of Calcium Silicate-Based Sealers and Epoxy Resin-Based Sealer on Human Periodontal Ligament Stem Cells

**DOI:** 10.3390/ma13225242

**Published:** 2020-11-20

**Authors:** Hanseul Oh, Egan Kim, Sukjoon Lee, Soyeon Park, Dongzi Chen, Su-Jung Shin, Euiseong Kim, Sunil Kim

**Affiliations:** 1Microscope Center, Department of Conservative Dentistry and Oral Science Research Center, Yonsei University College of Dentistry, 50-1 Yonsei-Ro, Seodaemun-Gu, Seoul 03722, Korea; ohs431@yuhs.ac (H.O.); egankim@yonsei.ac.kr (E.K.); shatoa@yuhs.ac (S.L.); dongzi-chen@yuhs.ac (D.C.); andyendo@yuhs.ac (E.K.); 2BK21 FOUR Project, Yonsei University College of Dentistry, 50-1 Yonsei-Ro, Seodaemun-Gu, Seoul 03722, Korea; sso88@yuhs.ac; 3Department of Conservative Dentistry, Gangnam Severance Dental Hospital, Yonsei University College of Dentistry, 211 Eonju-Ro, Gangnam-Gu, Seoul 06273, Korea; sujungshin@yuhs.ac; 4Department of Electrical & Electronic Engineering, Yonsei University College of Engineering, 50 Yonsei-Ro, Seodaemun-Gu, Seoul 03722, Korea

**Keywords:** calcium silicate-based sealer, biocompatibility, root canal treatment, epoxy resin-based sealer, cell viability, inflammatory response, periodontal ligament stem cell

## Abstract

The aim of this study was to evaluate the biocompatibility of calcium silicate-based sealers (CeraSeal and EndoSeal TCS) and epoxy resin-based sealer (AH-Plus) in terms of cell viability, inflammatory response, expression of mesenchymal phenotype, osteogenic potential, cell attachment, and morphology, of human periodontal ligament stem cells (hPDLSCs). hPDLSCs were acquired from the premolars (*n* = 4) of four subjects, whose ages extended from 16 to 24 years of age. Flow cytometry analysis showed stemness of hPDLSCs was maintained in all materials. In cell viability test, AH-Plus showed the lowest cell viability, and CeraSeal showed significantly higher cell viability than others. In ELISA test, AH-Plus showed higher expression of IL-6 and IL-8 than calcium silicate-based sealers. In an osteogenic potential test, AH-Plus showed a lower expression level than other material; however, EndoSeal TCS showed a better expression level than others. All experiments were repeated at least three times per cell line. Scanning electronic microscopy studies showed low degree of cell proliferation on AH-Plus, and high degree of cell proliferation on calcium silicate-based sealers. In this study, calcium silicate-based sealers appear to be more biocompatible and less cytotoxic than epoxy-resin based sealers.

## 1. Introduction

The purpose of root canal obturation is to form a tight barrier to protect the apical tissue from various microorganisms in the oral cavity [1]. The previous effort to form a tight barrier was to fill the root canal with gutta percha, provided that the gutta percha has better properties and biocompatibility than the sealer [1,2,3]. However, with the advent of calcium silicate-based sealers, the single-cone technique was recommended, and the amount of sealer entering the root canal was accordingly increased [4]. In addition, when clinicians inject calcium silicate-based sealers directly using a syringe needle in the root canal, there is a higher probability that the sealer will extrude beyond the root canal space compared with conventional techniques [1,5]. If the sealers are extruded beyond the physiological apical foramen, the unset sealers can irritate periapical tissues [6]. This may affect bone metabolism and healing in periapical tissues [7]. Therefore, it is necessary to evaluate the biocompatibility of sealers for human periodontal ligament stem cells (hPDLSCs).

In our previous study [5], some calcium silicate-based sealers were compared with epoxy resin-based sealers, and the results confirmed that the calcium silicate-based sealers had better biocompatibility than epoxy resin-based sealers. With the development of new calcium silicate-based sealers, compositional change of sealers could alter their biological characteristics and bioactivity [8]. Therefore, a study to evaluate their biocompatibility is needed if a new calcium silicate-based sealer is introduced.

CeraSeal is composed of tricalcium silicates, dicalcium silicates, calcium aluminates, zirconium oxides, and thickening agents. The synthesized pure calcium silicate compound was used for CeraSeal. López-García et al. reported that CeraSeal displayed higher cell viability, cell attachment, cell migration rates, and ion release rates than Endoseal. Furthermore, CeraSeal exhibited significantly more gene expression and mineralization capacity than Endoseal [9]. EndoSeal TCS is a developed product from EndoSeal MTA (Maruchi, Wonju, Korea), consisting of tricalcium silicates, phyllosilicate minerals, zirconium oxides, and dimethyl sulphoxides. Pure tricalcium silicate was used for EndoSeal TCS without using the Portland cement, which is the main component of EndoSeal MTA. However, there are rare studies regarding the biocompatibility of these newly developed calcium silicate-based sealers. Therefore, the purpose of this study was to evaluate the biocompatibility of newly introduced calcium silicate-based sealers in terms of cell viability, inflammatory response, expression of mesenchymal phenotype, osteogenic potential, cell attachment, and morphology.

## 2. Materials and Methods

### 2.1. Isolation and Culture of hPDLSCs

hPDLSCs were acquired from the premolars (*n* = 4) of four subjects, whose ages ranged from 16 to 24 years. Teeth were extracted for an orthodontic treatment plan, and consent forms were acquired according to the guidelines of the ethics committee of the research institution (Institutional Review Board number: 2-2017-0009). hPDL tissue was obtained from the mid-third of the root and minced into small pieces. The pieces of hPDL tissue were washed multiple times with Dulbecco′s modified Eagle′s medium (DMEM; Gibco, Grand Island, NY, USA) containing 3% penicillin-streptomycin (Gibco, Grand Island, NY, USA). The tissues were cultured in DMEM with 15% fetal bovine serum (FBS; Gibco, Grand Island, NY, USA) and 1% penicillin-streptomycin. Migrated hPDLSCs from the explants were passed onto passage 2 by cell separation with trypsin (Gibco, Grand Island, NY, USA). The separated hPDLSCs were cultured in DMEM with 10% FBS (Gibco) and 1% penicillin-streptomycin (normal culture media) in 5% CO_2_ (Changshin scientific, Seoul, Korea) at 37 °C. hPDLSCs at passages 4 to 6 were used for all experiments.

### 2.2. Preparation of Fresh/Setting Material Extraction Medium

Two calcium silicate-based sealers (CeraSeal; Meta-Biomed, Cheongju, Chungcheongbuk-do, Korea; EndoSeal TCS; Maruchi, Wonju, Gangwon-do, Korea) and an epoxy resin-based sealer (AH-Plus; DentsplySirona, Tulsa, OK, USA) were used as experimental materials (Table 1).

To prepare a fresh material extraction medium (fresh medium), all unset experimental sealer was prepared using manufacturers’ instructions, 200 mg, was placed in the 50 mL conical tube (Corning; Tewksbury, MA, USA) and 10 mL culture medium which consisted of DMEM, 10% FBS, and 1% penicillin-streptomycin was added to each tube. The tubes were incubated at 37 °C in 5% CO_2_ for 24 h. Afterwards, the supernatant of the fresh material extraction medium was filtered with a 0.2-μm pore-size filter (Minisart; Sartorius Stedim Biotech, Goettingen, Germany).

To prepare the setting material extraction medium (setting medium), all sealers were mixed and prepared using manufacturers’ instructions and incubated at 37 °C in 100% relative humidity for 48 h to allow them to be set completely. Set materials were made in the form of discs (5-mm diameter and 2-mm thickness, 200-mg weight) created from tube-shaped Teflon molds under sterile conditions. Each set sample disc was stored in 10 mL culture medium which consisted of DMEM, 10% FBS, and 1% penicillin-streptomycin at 37 °C in 5% CO_2_ for 24 h. Afterwards, the extracts were filtered using 0.2-μm pore-size filter (Minisart).

### 2.3. Measurement of the Mesenchymal Phenotype by Flow Cytometry Analysis (FACS)

The expression of mesenchymal stem cell surface molecules was analyzed using flow cytometry (FACS verse; BD bioscience, Piscataway, NJ, USA) assay. hPDLSCs (1 × 10^5^ cells) were seeded in a 6-well plate, and then after 24 h, the medium was changed to the setting medium and incubated for 3 days and 7 days. The cells were treated with 0.05% Trypsin-EDTA, washed with phosphate buffer solution (PBS), and a single-cell suspension was made for FACS analysis. The CD markers antigen-antibody reaction in hPDLSCs was examined with BD Stemflow human mesenchymal stem cell (MSC) analysis kit (BD Biosciences, Piscataway, NJ, USA). Target surface markers include the typical MSC markers (CD90, CD105, and CD73) and haematopoietic markers (CD11b, CD19, CD34, CD45, and HLA-DR) [10]. The data were analyzed using FlowJo (BD Biosciences, Piscataway, NJ, USA).

### 2.4. Cell Viability Assay (CCK-8)

Cell viability tests were performed with fresh media and setting media using the cell counting kit-8 (CCK-8; Dojindo Molecular Technologies, Rockville, MD, USA) assay. The hPDLSCs (1 × 10^4^ cells per well) were seeded in a 96-well plate and exposed to fresh and setting media. After 1, 3, and 7 days, cell viability was assessed using CCK-8 according to the manufacturer’s guidelines. The absorbance at 450 nm was measured using a spectrophotometer (Thermo Fisher Scientific, Waltham, MA, USA). The control group (normal hPDLSCs) and the experimental group were cultured in normal media at the same time and were evaluated.

### 2.5. Evaluation of Inflammatory Response by Enzyme-Linked Immunosorbent Assay (ELISA)

ELISA was performed to examine the inflammatory response of the materials. hPDLSCs (1 × 10^5^ cells) were seeded in a 6-well plate, and after 24 h, the medium was changed to fresh or setting medium and incubated for 24 h. IL-6 and IL-8 were examined as pro-inflammatory cytokines, and TGF-β as an anti-inflammatory cytokine [11]. The concentrations of IL-6, IL-8, and TGF-β in the fresh extraction medium were analyzed using an ELISA kit according to the manufacturer’s guidelines (R&D Systems, Minneapolis, MN, USA) and measured using a microplate reader (BioTek, Winooski, VT, USA) at 450 nm. The control group (normal hPDLSCs) and the experimental group were cultured in normal media at the same time and were evaluated.

### 2.6. Quantification of Osteogenic Potentials by Real-Time Quantitative Polymerase Chain Reaction (RT-qPCR)

hPDLSCs were cultured for 3 and 7 days in a setting media containing osteogenic-inducing reagents (10 mM β-glycerol phosphate, 100 µM l-ascorbic acid 2-phosphate, and 10 nM dexamethasone) to examine the pattern of osteogenic markers for all sealers using RT-qPCR. To examine the effects of sealers on osteoblastic differentiation, the mRNA expression levels of alkaline phosphatase (ALP), runt-related transcription factor 2 (RUNX2) and osteocalcin (OCN) were examined using the β-actin gene as an endogenous control [12]. Isolation of mRNA and transcription into cDNA was performed using the RNeasy mini kit (Qiagen, Hilden, Germany) and RevertAid first strand cDNA synthesis kit (Thermo Fisher Scientific, Waltham, MA, USA). qPCR was performed using the TaqMan gene expression assays (RUNX2, Hs01047943_m1; ALP, Hs00758162_m1; OCN; Hs01587814_g1, and β-actin, Hs01060665_g1) with the QuantStudio 3 RT-qPCR systems (Applied Biosystems, Foster City, CA, USA). The gene expression levels of ALP, RUNX2 and OCN were analyzed using this method. The control group (osteoinducted hPDLSCs) and the experimental group were cultured in an osteoinduction media (MEM + osteogenic-inducing reagents) at the same time and were evaluated.

### 2.7. Alizarin Red S (ARS) and Alkaline Phosphatase (ALP) Staining

ARS staining and ALP staining were conducted to examine the osteogenic activity. For ARS staining, hPDLSCs (2.5 × 10^4^ cells) were incubated for 14 days in the setting media containing osteogenic-inducing reagents. After 14 days, cells were fixed with 4% PFA, washed in PBS, and stained with 2% ARS solution (Acros, Gyeonggi-do, Korea). For ALP staining, hPDLSCs (2.5 × 10^4^ cells) were incubated for 3 and 7 days in the setting media including osteogenic-inducing reagents. After that, cells were fixed with 4% paraformaldehyde (PFA), washed in PBS, and stained using the ALP staining kit (Merck, Darmstadt, Germany). The ARS- and ALP-stained specimens were photographed using an optical microscope (Koptic, Gyeonggi-do, Korea), and digital camera (Nikon, Tokyo, Japan). The negative control groups (normal hPDLSCs) were cultured in normal media, while the positive control groups (osteoinducted hPDLSCs) were cultured in osteoinduction media at the same time as the experimental groups. These were stained and evaluated using the same protocol as above.

### 2.8. Evaluation of Cell Attachment and Material Surface Morphology by Scanning Electronic Microscopy

All set materials were made in the form of discs (5-mm diameter and 2-mm thickness, 200-mg weight) and incubated for 48 h to allow them to be set completely in an incubator (37 °C, 100% relative humidity, Changshin scientific, Seoul, Korea). After setting for 48 h, hPDLSCs (5 × 10^4^ cells/mL) were seeded onto each set disc directly and incubated at 37 °C in 5% CO_2_ for 72 h. The cultured discs were washed with PBS and fixed with 2% glutaraldehyde (Sigma-Aldrich, St. Louis, MO, USA). All discs were dehydrated, air-dried, and coated with 100-nm-thick Au/Pd to examine the cell attachment and morphology using field emission scanning electron microscopy (FE-SEM; Merlin, Carl Zeiss, Oerzen, Germany). Material surface morphology was examined after 48 h of material setting without hPDLSCs seeding in the same protocol as above.

### 2.9. Statistical Analysis

All experiments were repeated three times per cell line (*n* = 4). Differences among material groups were evaluated using the IBM SPSS statistics 25.0 software (IBM Corp., Armonk, NY, USA) with one-way analysis of variance followed by the Tukey’s test. Statistical significance was set at *p* < 0.05.

## 3. Results

### 3.1. Measurement of the Mesenchymal Phenotype by Flow Cytometry Analysis (FACS)

Throughout all experimental groups, including AH-Plus, the expression of the typical MSC markers (CD90, CD105, and CD73) was high (>99%). Hematopoietic markers (CD11b, CD19, CD34, CD45, and HLA-DR) showed low expression levels (<1%) (Figure 1).

### 3.2. Cell Viability Assay (CCK-8)

In fresh media, AH-Plus showed the lowest cell viability in all experimental periods. Calcium silicate-based sealers showed similar viability to control (cultured in normal media) at days 1 and 3, whereas at day 7, cell viability of CeraSeal significantly increased compared to control and EndoSeal TCS (Figure 2A). In setting media, on the other hand, cell viability of hPDLSCs was not significantly different between materials over all periods (Figure 2B).

### 3.3. Evaluation of Inflammatory Response by ELISA

As shown in Figure 3, the expression levels of pro-inflammatory cytokines (IL-6 and IL-8) were significantly higher in AH-plus than in other sealers, except for IL-6 in the setting media. Calcium silicate-based sealers showed statistically similar levels of IL-6 and IL-8 compared to the control (cultured in normal media) (Figure 3A–D). In the anti-inflammatory cytokine TGF-β, the expression level of all sealers in the fresh media was significantly lower than that in the control (Figure 3E). In the setting media, however, AH-Plus showed significantly lower expression levels of TGF-β than control and CeraSeal, and EndoSeal TCS was significantly lower than the control. CeraSeal remained at statistically similar levels to the control (Figure 3F).

### 3.4. Evaluation of Osteogenic Potentials by RT-qPCR and Staining

On day 3, all materials showed no significant differences observed in ALP, OCN, and RUNX2 expression (Figure 4). On day 7, ALP expression was significantly lower in AH-Plus than in other materials (Figure 4A). In OCN expression on day 7, EndoSeal TCS showed significantly higher levels than those of other materials, and AH-Plus showed a lower level than that of the control (cultured in osteoinduction media) (Figure 4B). In RUNX2 expression on day 7, the EndoSeal TCS showed a significantly higher level than that of the control, but there was no significant difference between the other materials (Figure 4C).

ALP staining (performed after 3 and 7 days) and ARS staining (performed after 14 days) are shown in Figure 5. In ALP staining on day 3, all materials were not stained enough to evaluate the difference. On day 7, ALP staining and ARS staining on day 14 showed that AH-Plus was less stained than calcium silicate-based sealers. CeraSeal and EndoSeal TCS showed similar ALP and ARS staining intensity to that of the positive control (cultured in osteoinduction media).

### 3.5. Evaluation of Cell Attachment and Material Surface Morphology by SEM

No cell adhesion was observed on the surface of AH-Plus, and several dead cells and their debris were observed (Figure 6A). On the surface of AH-Plus, AH-Plus showed a resinous matrix and fine filler particles (Figure 6B). The surface of the two calcium silicate set discs (CeraSeal and EndoSeal TCS) showed well-adhered hPDLSCs with the production of extracellular matrix and a high degree of cell proliferation (Figure 6C,E). By looking at the material surface next to the cells in CeraSeal, characteristic cubic particles can be identified (Figure 6C). On the surfaces of CeraSeal and EndoSeal TCS, honeycomb morphology or acicular spherule morphology was observed (Figure 6D,F).

## 4. Discussion

In this study, each experiment was conducted in fresh media or setting media. Experiments in fresh media were conducted to evaluate the initial response by the unset sealer after root canal filling. In fresh media, cell viability and inflammatory responses were measured; therefore, the initial cell toxicity and initial inflammatory response were evaluated. Experiments in setting media were conducted to evaluate the long-term response of the fully set sealer. In the setting media, mesenchymal phenotype measurement, cell viability test, inflammatory response measurement, and osteogenic potential evaluation were performed. Thus, the long-term responses of each aspect were evaluated.

hPDLSCs are a subpopulation of multipotent MSCs. We incubated hPDLSCs in setting media for 3 and 7 days to see how each set sealer affects the stemness of hPDLSCs. The stemness of cultured cells was evaluated using MSC criteria [9], and the MSC markers showed high levels (>99%), while the hematopoietic markers showed low levels (<1%) in both calcium silicate-based sealers and AH-Plus. This met the minimal criteria for defining MSCs [10]: MSC marker expression level ≥ 95% and hematopoietic marker expression level ≤2%. Therefore, it can be confirmed that the stemness of hPDLSCs was well maintained in a set medium of calcium silicate-based sealers and even AH-Plus.

Cell viability (CCK-8 assay) and inflammatory response tests (ELISA) were performed separately in fresh media and setting media. When the sealer is injected into the root canal, the fresh, unset sealer reacts with the periapical tissue. At this time, the viability or immune response of the cell contacting the sealers is directly related to postoperative pain and initial toxicity [5]. In fresh media, AH-Plus showed very high cytotoxicity all the time. The unset AH-Plus showed severe initial cytotoxicity in previous studies [5,13,14], which is thought to be due to the toxicity of the epoxy resin, one of the major components of AH-Plus [2]. Calcium silicate-based sealers showed similar cell viability to that of the control (cultured in normal culture media) at all times in fresh media. Notably, cell viability of CeraSeal significantly increased at day 7 compared to that of other groups, including the control. These results correspond with previous study assessing CeraSeal [9]. These results suggest that calcium silicate-based sealers have good biocompatibility and, in some cases, have the potential to promote cell growth.

Cell viability in the media was not significantly different among the AH-Plus, CeraSeal, EndoSeal TCS, and control groups in all experimental periods. AH-Plus, which showed high toxicity in fresh media, showed similar cell viability to that of the control after fully set. These results were consistent with the previous study of AH-Plus, which showed initial toxicity but became well tolerated within a few weeks [1]. Calcium silicate-based sealers were confirmed to have low cytotoxicity both in fresh media and in setting media.

The inflammatory response of hPDLSCs to the material was evaluated using ELISA, and the target cytokines were IL-6, IL-8, and TGF-β. IL-6 and IL-8 are pro-inflammatory cytokines that upregulate the inflammatory response and can be associated with inflammatory responses such as postoperative pain after canal obturation [15]. AH-Plus showed significantly higher pro-inflammatory cytokine levels (IL-6 and IL-8) than those of calcium silicate-based sealers, which is related to AH-Plus showing high toxicity in cell viability tests [16]. The pro-inflammatory cytokine levels of the two calcium silicate-based sealers were not different from those of the control (cultured in normal culture media), and this result also represented one aspect of their excellent biocompatibility. TGF-β is an anti-inflammatory cytokine, and its presence is known to contribute to the healing of apical periodontitis [11]. All the experimental groups showed lower levels of TGF-β than the level of control in fresh media. In setting media, AH-Plus and EndoSeal TCS showed lower levels of TGF-β than the control. CeraSeal showed higher levels of TGF-β than AH-Plus, and there was no significant difference between CeraSeal and control. Based on these results, it is possible to predict the anti-inflammatory effect of CeraSeal.

The osteogenic potential of the materials was quantified with RT-qPCR for ALP, OCN, and RUNX2 and visually qualified with ALP staining and ARS staining. If the sealers have osteogenic potential, apical healing can be promoted after canal obturation, and even the extrusion of the sealer may not interfere with the bony healing. ALP, OCN, and RUNX2 are essential transcription factors for the osteogenic differentiation of MSCs [17]. The more each of these factors is expressed, the more actively each process of osteogenic differentiation occurs [12,18]. From the results of day 7 of ALP, an early osteogenic differentiation marker, AH-Plus, showed significantly lower ALP expression than the other markers. Other calcium silicate-based sealers were not significantly different from the control (cultured in osteoinduction media). From the results of day 7 of OCN, a late osteogenic differentiation marker, EndoSeal TCS showed significantly higher expression than those of AH-Plus and CeraSeal. AH-Plus showed a significantly lower level of OCN than that of the control, as in ALP. From the results of day 7 of RUNX2, there was no significant difference between the materials, but the EndoSeal TCS showed a significantly higher RUNX2 level than that of the control. Overall, the EndoSeal TCS showed higher osteogenic potential, which may be related to the components of the EndoSeal TCS. Both CeraSeal and EndoSeal TCS are based on calcium silicates. Among calcium silicates, CeraSeal is composed of a mixture of tricalcium silicates and dicalcium silicates, but the EndoSeal TCS is composed only of tricalcium silicates. Tricalcium silicate is known to produce three times the amount of calcium hydroxide per molecule compared to dicalcium silicate [19,20]. It is thought that the EndoSeal TCS produces more calcium hydroxide than other materials, thereby showing a high osteogenic potential [21]. Likewise, AH-Plus is thought to have poor osteogenic potential because it does not have this ability [22].

The results of ALP and ARS staining were similar to those of RT-qPCR. The ALP staining kit reacts with ALP to produce a purple product, and the ARS staining kit stains calcium deposits in red, allowing direct observation of calcium mineralization. In the ALP 7 days and ARS 14 days groups, CeraSeal and EndoSeal TCS showed similar staining in positive control (cultured in osteoinduction media), whereas AH-Plus stained less. Based on the results of the ALP, OCN, and RUNX2 mRNA expression tests and ALP and ARS staining tests, it is thought that the EndoSeal TCS can exhibit better osteogenic potential than other materials and further promote bony healing. CeraSeal shows a similar value to the control (cultured in osteoinduction media), so it is thought not to interfere with normal bony healing. The good osteogenic potential of calcium silicate sealers demonstrated in this study is consistent with previous studies [5,9]. In contrast, AH-Plus is thought to inhibit bony healing, so care should be taken to ensure that AH-Plus is not extruded out of the apical foramen [22].

Cell attachment and morphology of hPDLSCs on the set material disc were examined using the secondary electron mode of SEM. Since the additional process of culturing the cell itself may affect the crystal structure of the material [23], the surface structure was observed after 48 h of material setting without cell culture. When the cells were cultured on AH-Plus, all cells were observed to be dead in a round shape that hardly is differentiated (Figure 6A). This is the result corresponding to the high cytotoxicity of AH-Plus in the cell viability test of this study. Looking at the surface of the AH-Plus material (Figure 6B), the resinous matrix is on the ground, and filler particles, which are characteristics of AH-Plus that can be observed [24]. Unlike the epoxy resin sealer, cells on the surface of calcium silicate-based sealers proliferate well and differentiate well. Upon observing the cell growth on the CeraSeal surface (Figure 6C), hPDLSCs were well adhered to the surface of the material and appeared to be well differentiated, extending in all directions. On the cell surface, extracellular matrices with globular form and reticulum form are also observed. Inspecting the surface of the CeraSeal set for 48 h (Figure 6D), a honeycomb appearance or acicular spherule appearance, which is a typical crystal form of calcium silicate hydrate (C-S-H), was observed [25,26]. In addition, on the surface of CeraSeal with cells (Figure 6C), characteristic cubic forms were seen next to the cells. The form of its crystals appeared to be a typical form of C_3_AH_6_ (3CaO∙Al_2_O_3_∙6H_2_O, Hydrogarnet) [25,27]. C_3_A (Ca_3_Al_2_O_6_, tricalcium aluminate), one of the constituents of CeraSeal, usually reacts with gypsum (CaSO_4_, calcium sulphate) in usual MTA and forms needle-like ettringite (calcium sulphoaluminate). However, there are no such gypsum components in CeraSeal, so it is thought that cubic-shaped C_3_AH_6_ was formed [25]. This C_3_AH_6_ phase is known to increase the initial strength of MTA [25], and it is thought that the cubic crystals grew by hydration for an additional 72 h during cell culture. The cell growth on the EndoSeal TCS disc was active like CeraSeal, and the cells were well differentiated into characteristic elongated forms (Figure 6E). Since the cells directly contacted calcium silicate-based sealers, these cells grew much better than that of the epoxy resin-based sealer; therefore, it was confirmed that the biocompatibility of calcium silicate-based sealers in the cell level was superior. The crystal form of the EndoSeal TCS was observed with a typical honeycomb appearance or acicular spherule appearance (Figure 6F). Through this, it is considered that the initial crystalline phase of the EndoSeal TCS is calcium silicate hydrate (C-S-H), similar to CeraSeal. However, since this study has limitations as an in vitro study, future in vivo studies such as subcutaneous tissue implantation model or dog root canal filling model studies would be necessary to strengthen the rationale for the new materials and techniques [28,29].

## 5. Conclusions

According to this study, it was confirmed that calcium silicate-based sealers (CeraSeal, EndoSeal TCS) are less cytotoxic and more biocompatible than epoxy resin-based sealers. In particular, CeraSeal showed less cytotoxicity than the other materials before setting, and the EndoSeal TCS showed better osteogenic potential than the other materials. Future in vivo studies or clinical studies would be necessary to strengthen the rationale for the new materials.

## Figures and Tables

**Figure 1 materials-13-05242-f001:**
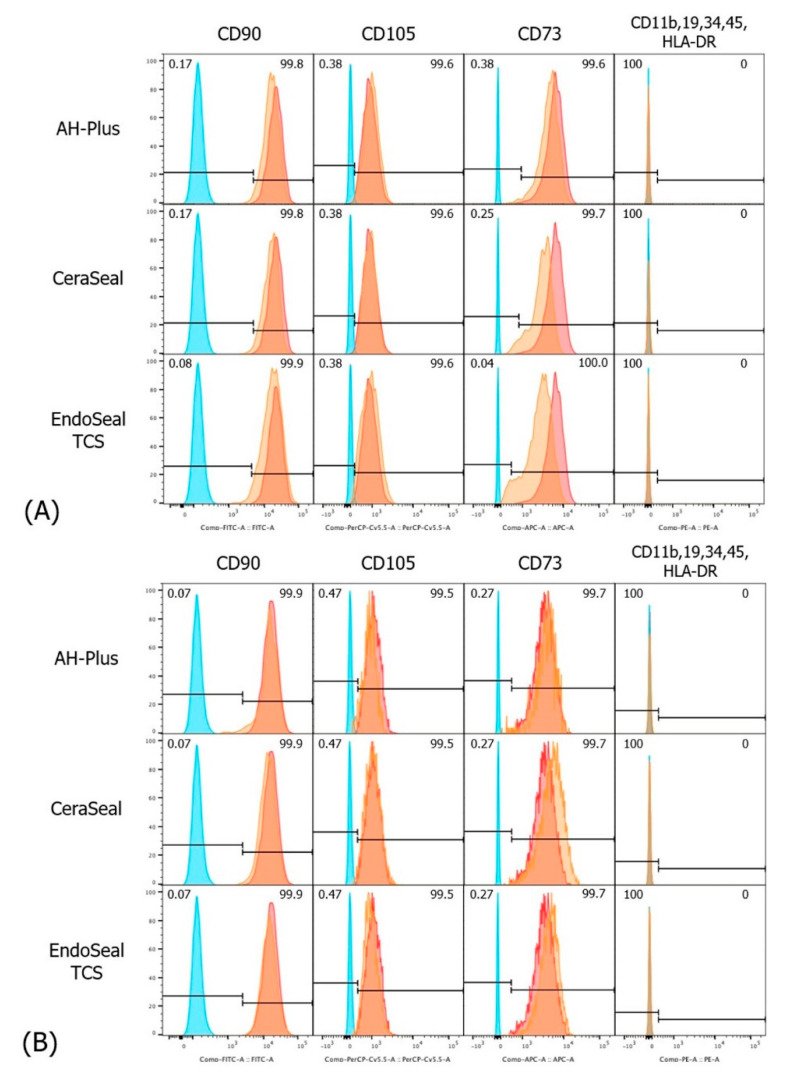
Mesenchymal phenotype expression of human periodontal ligament stem cells (hPDLSCs). (**A**) 3 days, (**B**) 7 days. Blue tracing, Isotype; Orange tracing, experimental group; Red tracing, normal group cultured in normal culture media.

**Figure 2 materials-13-05242-f002:**
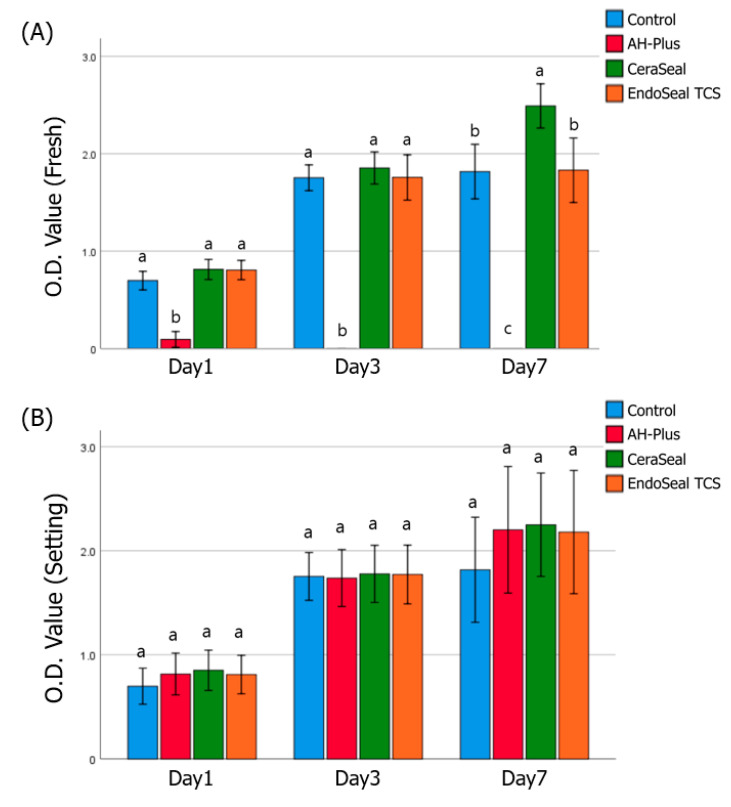
Cell viability of human periodontal ligament stem cells (hPDLSCs). (**A**) fresh sealers, (**B**) set sealers. Different alphabets indicate that there is a difference between the materials within the same time group (*p* < 0.05).

**Figure 3 materials-13-05242-f003:**
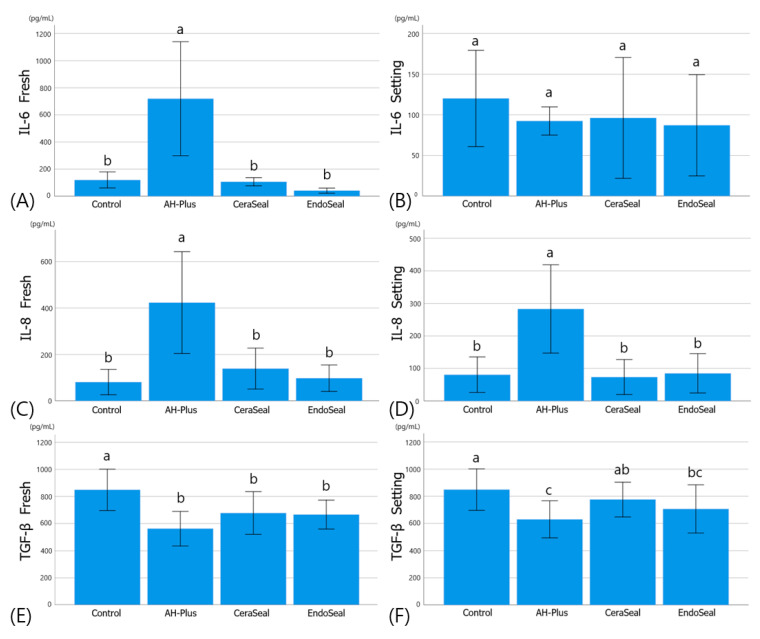
Inflammatory response of hPDLSCs. (**A**) IL-6 expression in fresh media, (**B**) IL-6 expression in setting media, (**C**) IL-8 expression in fresh media, (**D**) IL-8 expression in setting media, (**E**) TGF-β expression in fresh media, and (**F**) TGF-β expression in setting media. Different alphabets indicate that there is a difference between the materials within the same experimental group (*p* < 0.05).

**Figure 4 materials-13-05242-f004:**
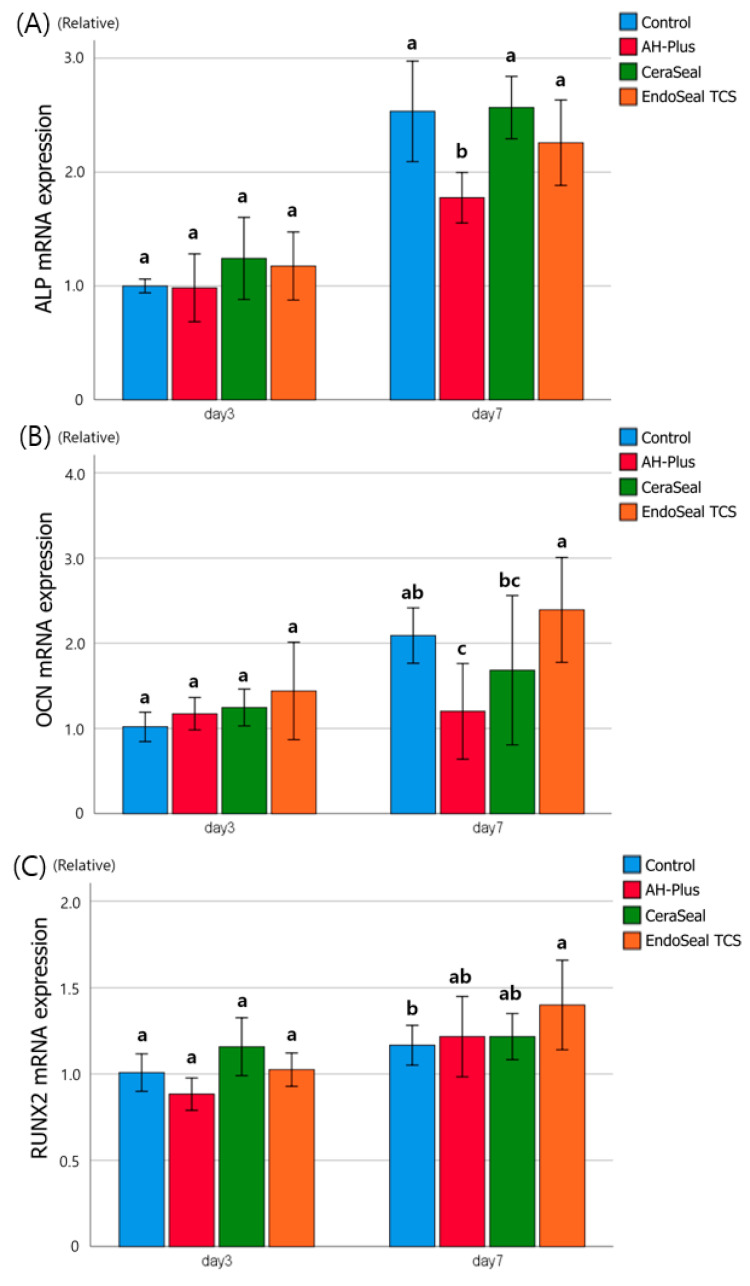
The effects of sealers of hPDLSCs on the relative mRNA expression. (**A**) ALP, (**B**) OCN, and (**C**) RUNX2 measured with RT-qPCR and normalized with β-actin. Different alphabets indicate that there is a difference between the materials within the same time group (*p* < 0.05).

**Figure 5 materials-13-05242-f005:**
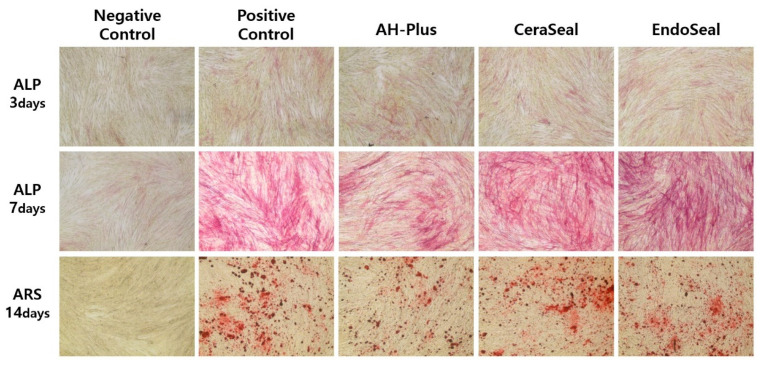
The effects of sealers on the osteogenic differentiation of hPDLSCs. The 40× magnified microscope images are shown. Negative control, cultured in normal culture media; Positive control, cultured in osteoinduction media (normal culture media with osteogenic-inducing reagents); Experimental groups, cultured in setting media with osteogenic-inducing reagents.

**Figure 6 materials-13-05242-f006:**
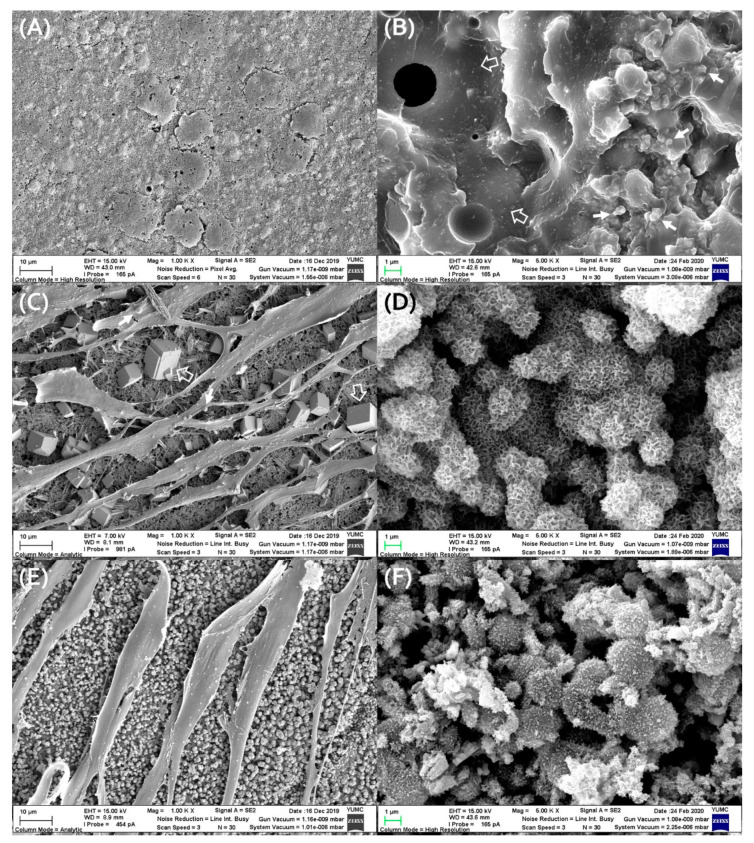
Cell attachment of hPDLSCs on the set material disc and surface of set material disc. Cell attachment (**A**,**C**,**E**) is shown at magnification of 1000× and material surface morphology (**B**,**D**,**F**) is shown at magnification of 5000×. (**A**) Cell attachment and morphology on the set AH-Plus. (**B**) Surface morphology of the set AH-Plus; AH-Plus showed resinous matrix (open arrow) and fine filler particles (closed arrow). (**C**) Cell attachment and morphology on the set CeraSeal; Well-adhered hPDLSCs with production of globular extracellular matrix (closed arrow) were shown and characteristic cubic particles (open arrow) can be identified. (**D**) Surface morphology of the set CeraSeal; Honeycomb morphology or acicular spherule morphology is observed. (**E**) Cell attachment and morphology on the set EndoSeal TCS; High degree of cell proliferation and well-adhered hPDLSCs were shown. (**F**) Surface morphology of the set EndoSeal TCS; Typical acicular spherule crystals are observed.

**Table 1 materials-13-05242-t001:** The information of the tested sealers in this study. AH-Plus contains bisphenol A, which is a major component of epoxy-resin. CeraSeal and EndoSeal TCS contain several calcium silicates.

Sealer	Manufacturer	Composition	
AH-Plus	DentsplySirona, Tulsa, OK, USA	Paste A 25–50% bisphenol A 10–25% zirconium dioxide NS calcium tungstate NS iron oxide	Paste B 2.5–10% N, n-dibenzyl-5-oxanonandiamin-1,9 2.5–10% amantadine
CeraSeal	MetaBiomed, Cheongju, Korea	Tricalcium silicate, Dicalcium silicate, Calcium aluminate, Zirconium oxide, Thickening agent	
EndoSeal TCS	MARUCHI, Wonju, Korea	Tricalcium silicate, Phyllosilicate mineral, Zirconium oxide, Dimethyl sulfoxide	

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
