# Peer review of "Comparison of Biocompatibility of Calcium Silicate-Based Sealers and Epoxy Resin-Based Sealer on Human Periodontal Ligament Stem Cells"

_materials, 2020, doi:10.3390/ma13225242_

Round 1

Reviewer 1 Report

Dear Editor,

After the review process, I have several comments: you should insert numerical data in the abstract; you should insert references in all Materials and Methods; you should insert limitations and future perspectives of the study; you should rewrite the conclusions section.

Best regards!

Reviewer 2 Report

This research assesses in vitro the biocompatibility of 2 novel calcium-silicate based sealers on human periodontal ligament stem cells.

The study is under the scope of the journal. Authors choose AH Plus as the control group sealer, which is accepted among endodontics community. The research is well designed and the battery of tests to assess biocompatibility and bioactivity are according to the current best practices.  Study methods are valid, reliable and presented with sufficient detail. Results are presented correctly and with good level of detail.

However, some issues need to be solved:

The introduction presents the gap in knowledge mostly in a correct way, nevertheless, authors most acknowledge the existence of a previous study assessing Ceraseal, and appropriately report on their findings:

https://doi.org/10.1007/s00784-019-03036-2 .

Discussion of the results needs to be deeper, thoroughly comparing the results of the present study with the previous Ceraseal study and other calcium silicate-based sealers biocompatibility and bioactivity assessment using similar methodology.

Moreover, the limitations of evaluating in vitro the biocompatibility of root canal sealers most be presented, and in vivo models for assessing the biocompatibility profile of the materials must be presented, as subcutaneous

 https://doi.org/10.2334/josnusd.18-0145 or intraosseous implantation.

Also, other models to explore the biocompatibility and functional efficacy of root canal sealers to have a better insight of periapical tissue reaction towards the newly purposed materials J Endod. 2014 Jun;40(6):837-41. doi: 10.1016/j.joen.2013.10.023 may add to improve the paper in order to give a wider view of biological evaluation of root canal sealers.

P11-L289 interrogated? Please consider other word.

Bibliography is up-to-date and well cited, however, ref 18 seems to miss authors names.

Reviewer 3 Report

The authors have prepared a well documented and well driven paper on the topic of biocompatibility of calcium silicate-based sealers and epoxy resin-based sealers. Various parameters were measured to determine cell viability, inflammatory response, and other factors. The paper is well balanced between sections and very eloquent, with a good writing style. However, the manuscript could further be improved, in this reviewer's opinion, as follows:

  • Why was one-way analysis of variance preferred over two-way analysis in the comparison between groups? Were changes expected to occur only in one direction and why?
  • Figure 2 shows asterisc as p < 0.05 , and in subfigure A it is drawn in all time periods of AH Plus but also in CeraSeal on day 7. This should be clarified as it is unclear what the significant difference/correlation is. Similarly, for Figure 3 A,C,D,E, and 4 A.
  • The authors state that material surface morphology was examined via electron microscopy after 48 h of material setting. However, the timestamps on the images show a difference of at least 2 months from the cell attachment images. How do you explain this? Is the time interval a factor? How were the discs kept? Could they have deteriorated in some way, or have been exposed to aggressive factors? Is keeping them in a different environment than the oral cavity of influence in the results?
  • Table 1 should clarify that epoxy resin is a major component of AH Plus, since this represents a base for discussions.
  • At least one study citing the osteogenic potential and effect on bony healing of AH-Plus should be added, as these actions are mentioned repeatedly without references.

That said, the paper has great potential and could be of interest to the readers of this Journal.

Respectfully submitted,

Reviewer 4 Report

The manuscript presents the comparison of biocompatibility of two calcium silicate-based sealers and an epoxy resin-based sealer. Majority of the cell-experiments are based on the extract medium for the unset and set sealants. The manuscript is well written and presents good experimental data. I have, however, few minor concerns which I think are important to improve overall quality of results presented.

  1. How was the extract media concentration of 20 mg/ml selected? Were there any standards that were followed for cytotoxicity testing?

  1. What was the size of the particles added to prepare the Fresh medium? How was the filter size of 0.2 µm selected? I would like the authors to slightly expand the method section 2.2 explaining these as the whole study is based on the extracted media.

  1. Were the ELISA results for pro-inflammatory markers normalized to the cell number? When we look at the CCK-8 results for fresh medium on figure 2, OD value for AH-PLUS group is significantly low, however, the expression of IL-6 and IL-8 is significantly higher than other groups.

  1. Even more interestingly, between Fresh and Setting media, for AH-PLUS group there is a big difference in OD value (Fresh: close to 0 and Setting: close to 1), however, the concentration of IL-8 does not look that different. I would like the authors to clarify this and revise if necessary. I think normalization with cell number might be necessary.
